# Control and regulation of acetate overflow in *Escherichia coli*

**Pierre Millard[1,2]\*, Brice Enjalbert[1], Sandrine Uttenweiler-Joseph[1], Jean-Charles Portais[1,2,3], Fabien Létisse[1,4]**

[1]TBI, Université de Toulouse, CNRS, INRAE, INSA, Toulouse, France; [2]MetaToul-MetaboHUB, National Infrastructure of Metabolomics and Fluxomics, Toulouse, France; [3]RESTORE, Université de Toulouse, INSERM U1031, CNRS 5070, Université Toulouse III - Paul Sabatier, EFS, Toulouse, France; [4]Université Toulouse III - Paul Sabatier, Toulouse, France

**Abstract** Overflow metabolism refers to the production of seemingly wasteful by-products by cells during growth on glucose even when oxygen is abundant. Two theories have been proposed to explain acetate overflow in *Escherichia coli* – global control of the central metabolism and local control of the acetate pathway – but neither accounts for all observations. Here, we develop a kinetic model of *E. coli* metabolism that quantitatively accounts for observed behaviours and successfully predicts the response of *E. coli* to new perturbations. We reconcile these theories and clarify the origin, control, and regulation of the acetate flux. We also find that, in turns, acetate regulates glucose metabolism by coordinating the expression of glycolytic and TCA genes. Acetate should not be considered a wasteful end-product since it is also a co-substrate and a global regulator of glucose metabolism in *E. coli*. This has broad implications for our understanding of overflow metabolism.

\*For correspondence:
millard@insa-toulouse.fr

**Competing interests:** The authors declare that no competing interests exist.

## Introduction

Overflow metabolism refers to the production of seemingly wasteful by-products by cells during growth on glycolytic substrates such as glucose, even when oxygen is abundant. Overflow metabolism has been reported for most (micro)organisms: yeasts produce ethanol (a phenomenon known as the Crabtree effect; *Crabtree, 1929*), mammalian cells produce lactate (the Warburg effect in cancer cells; *Warburg, 1956*), and bacteria produce small organic acids such as acetate (*Harden, 1901*). This ubiquitous phenomenon has been extensively investigated because of its fundamental and applied importance, but its origin and regulation remain to be clarified.

Acetate production by *Escherichia coli* has been studied for decades as a model of overflow metabolism. The recent rise of systems biology, which combines experimental techniques and mathematical modelling to achieve a mechanistic understanding of processes underlying the function of biological systems (*Bruggeman et al., 2007*; *Shou et al., 2015*), has provided a quantitative understanding of the cause of acetate overflow in *E. coli*. Acetate production is considered to result from an imbalance between *E. coli*'s capacities to produce and assimilate acetyl-CoA, the main precursor of acetate. Stoichiometric models suggest that this imbalance could be caused by various cell-level constraints, such as energy conservation (*de Groot et al., 2020*; *El-Mansi and Holms, 1989*), recycling of cofactors (*Wolfe, 2005*; *Vemuri et al., 2006*), membrane occupancy (*Zhuang et al., 2011*; *Szenk et al., 2017*), and resource allocation (*Basan et al., 2015*; *Zeng and Yang, 2019*). These models – and the underlying theories – capture the essence of the overflow process and successfully predict the growth rate dependence of acetate production in *E. coli* (*Basan et al., 2015*; *Zeng and Yang, 2019*; *Renilla et al., 2012*).

However, none of these theories explain why increases in extracellular acetate concentration can interrupt or even reverse the acetate flux in *E. coli* (*Enjalbert et al., 2017*), which is able to co-consume glucose and acetate even when glucose is abundant. Cell growth is not affected when acetate production is abolished (*Enjalbert et al., 2017*; *Lee and Liao, 2008*), suggesting that this overflow is neither necessary for *E. coli* to sustain fast growth nor strictly imposed by the proposed constraints. This phenomenon cannot be reproduced by stoichiometric models since they do not account for metabolite concentrations. Kinetic models are therefore a necessary step towards a comprehensive understanding of the dynamics, control, and regulation of metabolic systems. For example, a kinetic model of the Pta-AckA pathway has successfully been used to predict the reversal of the acetate flux at high acetate concentrations (*Enjalbert et al., 2017*), a significant advance compared to stoichiometric models of acetate overflow. This effect is caused by thermodynamic control of the acetate pathway by the concentration of acetate itself, a mechanism that operates independently of enzyme expression. Still, thermodynamic control does not imply that enzymes exert no control whatsoever over acetate flux, though the degree of this control remains to be clarified. Moreover, this local mechanism does not explain the toxic effect of acetate on microbial growth (*Wilbanks and Trinh, 2017*; *Luli and Strohl, 1990*).

In this work, we used a systems biology approach to clarify the origin, control, and regulation of acetate overflow in *E. coli* over the broad range of acetate concentrations this bacterium experiences under laboratory and industrial conditions as well as in its environmental niche (*Wolfe, 2005*; *Shen et al., 2012*; *de Graaf et al., 2010*; *Kleman and Strohl, 1994*; *Jones et al., 2007*; *Cummings et al., 2001*; *Macfarlane et al., 1992*; *Cummings and Englyst, 1987*; *Fabich et al., 2008*).

## Results

### Construction of a kinetic model of *E. coli* metabolism

Following a top-down systems biology approach (*Bruggeman et al., 2007*), we constructed a coarse-grained kinetic model of *E. coli* metabolism that links acetate metabolism with glucose uptake and growth (*Figure 1A*). This model includes three processes: (i) glucose transport and its conversion into acetyl-CoA by the glycolytic pathway, (ii) utilisation of acetyl-CoA in the TCA cycle (energy conservation) and anabolic pathways (production of building blocks) for growth, and (iii) acetate metabolism, i.e., the reversible conversion of acetyl-CoA into acetate via phosphotransacetylase (Pta), acetate kinase (AckA), and an acetate exchange reaction between the cell and its environment.

This initial model included two compartments, 6 species, 6 reactions, and 24 parameters. The value of 14 parameters were taken directly from the literature (*Enjalbert et al., 2017*; *Millard et al., 2017*; *Kadir et al., 2010*). To estimate the remaining parameters, we performed three growth experiments on $^{13}$C-glucose (15 mM) plus different concentrations of $^{12}$C-acetate (1, 10, and 30 mM), as detailed in *Enjalbert et al., 2017*. These experiments were designed to demonstrate that *E. coli* simultaneously produces acetate from glucose and consumes it from the environment (*Enjalbert et al., 2017*). They provide information on the forward and reverse fluxes of acetate between the cell and its environment (*Enjalbert et al., 2017*), and thereby improve the calibration of the model. The model was extended with isotopic equations to account for the propagation of $^{13}$C label (*Enjalbert et al., 2017*; *Millard et al., 2015*), and parameters were estimated by fitting concentration time courses of glucose, biomass, and acetate and $^{13}$C-enrichment of acetate (see Materials and methods). This initial model (model 1) did not fit the data satisfactorily (*Figure 1B*, *Figure 1—figure supplement 1*, Materials and methods). Adding inhibition of the glycolytic pathway by acetate (model 2) to account for the reduction in glucose uptake at high acetate concentrations improved the fits of the biomass, glucose, and acetate concentration profiles but not of the acetate labelling profiles (*Figure 1—figure supplement 2*). Similarly, adding inhibition of the TCA cycle by acetate (model 3) slightly improved the fit of the acetate labelling profile but not of the glucose and biomass concentration profiles (*Figure 1—figure supplement 3*). A sufficiently accurate fit was only achieved when both pathways were inhibited (model 4; *Figure 1B,C*). These modifications suggest the existence of an unknown regulatory program that makes *E. coli* sensitive and responsive to acetate concentration, and provide a mechanistic rationale for the reported 'toxicity' of acetate to *E. coli* (*Enjalbert et al., 2017*; *Wilbanks and Trinh, 2017*; *Luli and Strohl, 1990*; *Pinhal et al., 2019*).

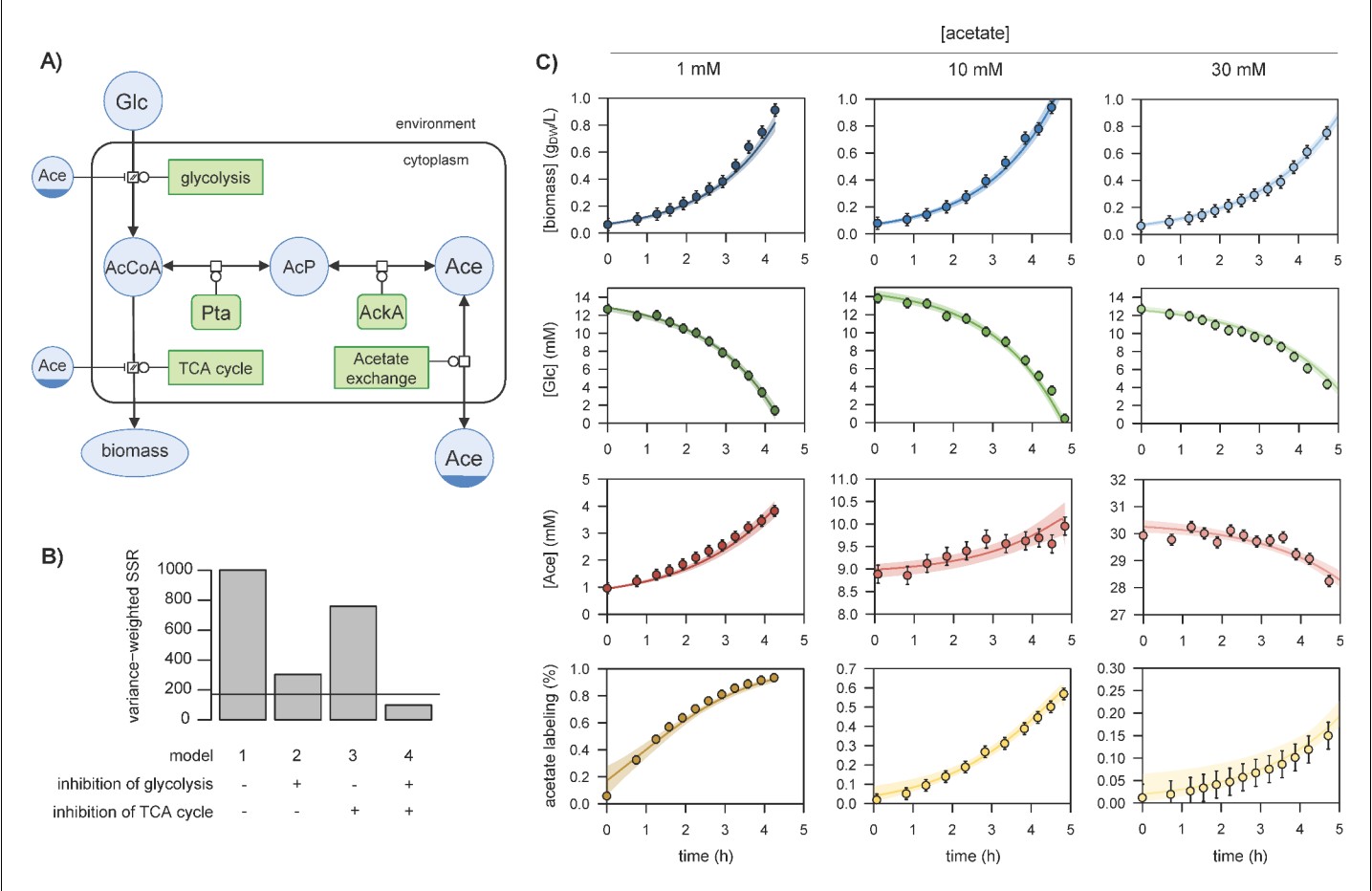

**Figure 1.** Representation of glucose and acetate metabolism in *Escherichia coli* (A), in Systems Biology Graphical Notation format (http://sbgn.org) (*Le Novère et al., 2009*). We performed [13]C-labelling experiments to calibrate the model and evaluated the goodness of fit for different topologies (B). The initial model (model 1), which does not include inhibition of the glycolytic pathway and TCA cycle by acetate, did not fit the data satisfactorily. Adding inhibition by acetate of glycolysis (model 2) or of the TCA cycle (model 3) improved the fit, but both pathways had to be inhibited (model 4) for the goodness-of-fit criterion to be satisfied. In (B), the horizontal line represents the 95% confidence threshold for the variance-weighted sum of squared residuals (SSR). The best fits of the experimental data obtained with model 4 are shown in (C), where the shaded areas represent the 95% confidence interval on the fits. The best fits obtained with the alternative models (models 1–3) are shown in *Figure 1—figure supplements 1–3*. The online version of this article includes the following source data and figure supplement(s) for figure 1:

**Source data 1.** Experimental data used to calibrate the model.
**Figure supplement 1.** Best fit obtained for model 1.
**Figure supplement 2.** Best fit obtained for model 2.
**Figure supplement 3.** Best fit obtained for model 3.

## Acetate regulates glucose uptake, glycolysis, and the TCA cycle at the transcriptional level

A key hypothesis required for the model to fit the experimental data is that acetate inhibits the flux capacity of both the glycolytic pathway and the TCA cycle in *E. coli*. In this model, the aim of the simplified description of inhibition by acetate was to represent the integrated response of *E. coli* metabolism to acetate, with no a priori knowledge of the underlying molecular mechanism(s). To determine whether this inhibition actually occurs in vivo at the transcriptional level, we monitored gene expression in *E. coli* grown on glucose (15 mM) plus acetate at different concentrations (0, 10, 50, or 100 mM).

Transcriptomic results revealed a global and progressive remodelling of gene expression at increasing acetate concentrations (*Figure 2A–B*). We noted that the presence of acetate modulates the expression of genes involved in various biological functions: motility, biofilm formation,

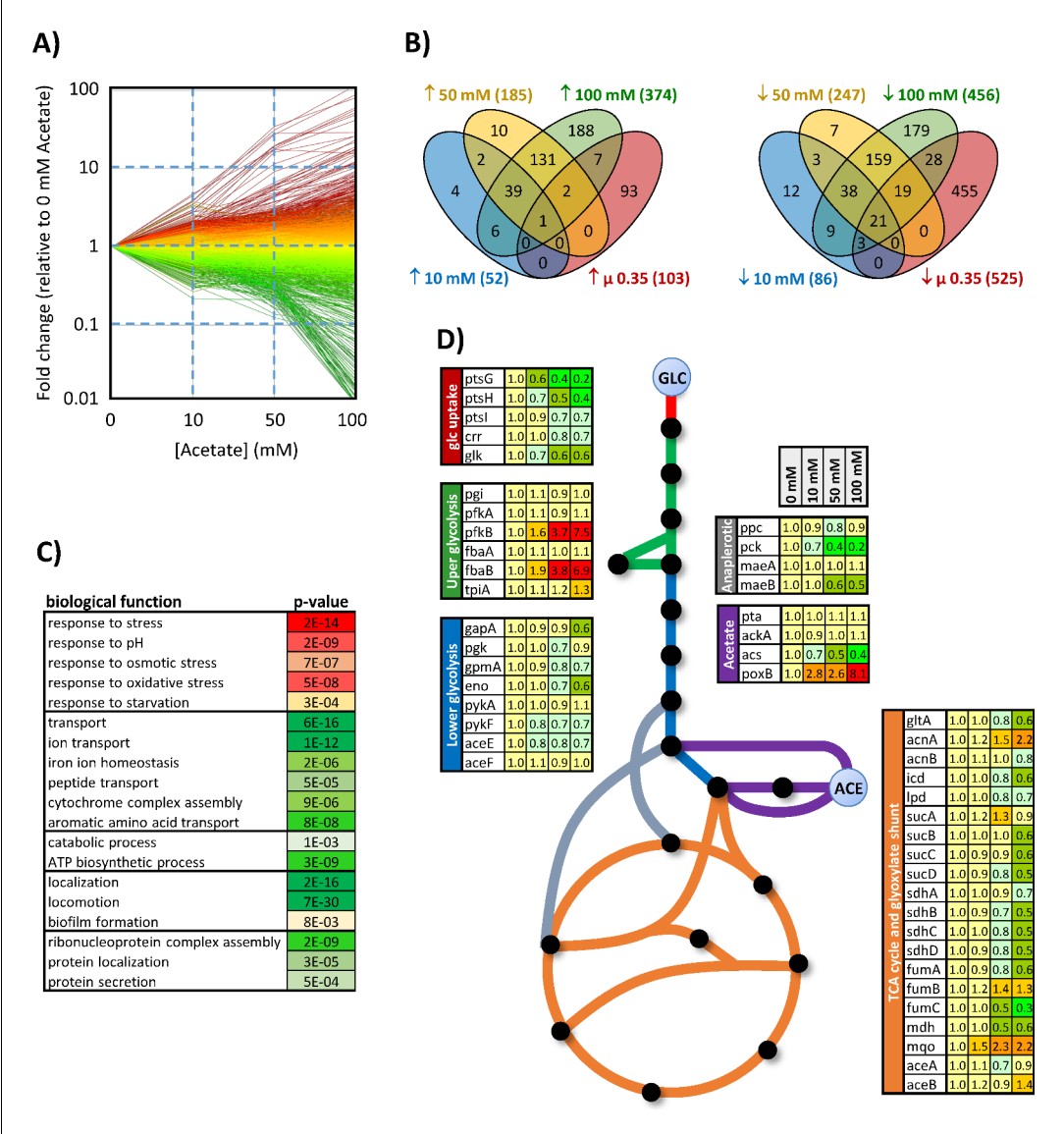

**Figure 2.** Response of the *E. coli* transcriptome to changes in acetate concentration (0, 10, 50, or 100 mM) during growth on glucose (15 mM). The changes in gene expression are shown in (**A**). Each line represents the expression of a single gene relative to its expression level measured in the absence of acetate. Up- and downregulated genes are shown in red and green, respectively. The Venn diagrams (**B**) represent the total number of genes upregulated (left) and downregulated (right) by at least a factor of 2 under each condition and during growth on glucose in the absence of acetate but at the same growth rate as in the presence of 100 mM acetate (0.35 hr$^{-1}$, extrapolated from the data from *Esquerré et al., 2014*). Biological functions modulated by the presence of acetate (based on Gene Ontology analysis) are shown in (**C**), with the corresponding p-values. The expression levels of central metabolic genes are shown in (**D**). The data were obtained from four independent biological replicates for each condition. The online version of this article includes the following source data for figure 2:

**Source data 1.** Response of the *E. coli* transcriptome to changes in acetate concentration (0, 10, 50, or 100 mM) during growth on glucose (15 mM).

translation, stress response, metabolism, transport of carbohydrates, amino acids, and ions (*Figure 2C*). Importantly, the observed changes in gene expression cannot result solely from the inhibition of growth by acetate, as shown by comparing gene expression levels on glucose plus 100 mM acetate with those on glucose alone at the same growth rate (0.35 hr$^{-1}$) (*Figure 2B*).

At the level of the central metabolism (*Figure 2D*), acetate reduces the expression of all genes that code for the glucose phosphotransferase system (PTS) (*ptsGHI*, *crr*), without inducing the expression of alternative systems of glucose internalisation and phosphorylation (*galP*, *mglABC*, *glf*, *glk*, *manXYZ*) that could have compensated for reduced PTS activity and the corresponding

inhibition of glucose uptake (*Lim and Jung, 2017*; *Ruyter et al., 1991*). Expression of upper glycolytic genes remained stable, with the exception of two isoenzymes that were overexpressed in the presence of acetate: *fbaB*, a gluconeogenic enzyme, and *pfkB*, which contributes little to phosphofructokinase activity on glucose (<10%) (*Fraenkel, 1986*; *Scamuffa and Caprioli, 1980*; *Long and Antoniewicz, 2019*; *Daldal et al., 1982*). In contrast, the expression of most of the lower glycolysis genes (*pgk, gapA, gpmA, eno, pykF, aceE*) was reduced by 15–40% at 100 mM acetate. At this concentration, acetate also inhibits the expression of virtually all TCA cycle genes (*gltA, acnAB, icd, lpd, sucABCD, sdhABCD, fumABC, mdh*) by 30–67%. In terms of acetate metabolism, the expression of *pta* and *ackA*, which code for enzymes in the Pta-AckA pathway, remained remarkably stable at all acetate concentrations. Expression of pyruvate oxidase (*poxB*) was increased by a factor of 8 in the presence of acetate, though this promiscuous enzyme contributes to acetate metabolism mainly in the stationary phase and apparently not under our conditions (*Enjalbert et al., 2017*; *Pinhal et al., 2019*; *Martínez-Gómez et al., 2012*; *Dittrich et al., 2005*). Expression of acetyl-CoA synthetase (*acs*) – which converts acetate to acetyl-CoA with a high affinity – decreased with increasing acetate concentration, indicating that under glucose excess, the presence of acetate does not activate acetate recycling through the Pta-AckA-Acs cycle (*Enjalbert et al., 2017*; *Valgepea et al., 2010*).

These data confirm our hypothesis that acetate gradually modulates both the flux capacity of *E. coli* to produce acetyl-CoA from glucose and its capacity to utilise acetyl-CoA as a source of energy and of anabolic precursors for growth. Acetate does not appear to influence the flux capacity of the acetate pathway itself, which is also consistent with the proposed model.

## Testing the model

We tested the model by predicting the response of *E. coli* to new perturbations and comparing the results with experimental data (*Figure 3*).

First, we checked that the model could reproduce the established growth-rate dependence of acetate production in *E. coli*. We modified the model to simulate glucose-limited chemostat experiments (by adding reactions for glucose feed and medium outflow), and we predicted the steady-state glucose and acetate fluxes at different dilution rates (from 0.1 to 0.5 $hr^{-1}$). Since the present model cannot account for the activation of acetyl-CoA synthetase – involved in acetate assimilation – under glucose limitation (*Valgepea et al., 2010*), model simulations were compared to experimental data collected on a Δ*acs* strain (*Renilla et al., 2012*). The simulated profiles of acetate production as a function of growth and glucose uptake rates were in excellent agreement with experimental data (*Figure 3A*), indicating that the model accurately captures the effects of glucose limitation on growth and acetate fluxes.

Second, we evaluated the response of *E. coli* to changes in acetate concentration. We simulated steady-state acetate, glucose, and growth fluxes under glucose excess over a broad range of acetate concentrations (between 100 µM and 100 mM) (*Enjalbert et al., 2017*). The growth and glucose uptake rates decreased monotonously with increasing acetate concentrations, in agreement with the experimental data (*Figure 3B*). The acetate flux profile was also well described by the model, with a progressive decrease of acetate production with increasing acetate concentration, and a reversal of the acetate flux above a threshold concentration of about 10 mM. As observed experimentally, the growth rate was not affected when acetate production was abolished, confirming that acetate production is not required to maintain fast growth nor is it imposed by any intracellular constraint.

Third, we tested the dynamic properties of the model by predicting the response of *E. coli* to sudden variations in the acetate concentration, which is not possible using existing models. In the exponential growth phase on glucose, we simulated the time courses of the changes in the concentrations of glucose and acetate after a pulse of acetate (30 mM) or of water (control experiment), as described in *Enjalbert et al., 2017*. Here again, the model accurately reproduces the experimental profiles, in particular the rapid reversal of the acetate flux and reduction of glucose uptake after the acetate pulse (*Figure 3C*).

Overall, the predictions of the model are in agreement with experimental findings. The model accurately predicts the steady-state and dynamic relationships between glucose uptake, growth, and acetate fluxes in *E. coli* over a broad range of glucose and acetate concentrations. Importantly, the model was *not* trained on these data (i.e., they were not used to calibrate the model), so this test is a strong validation of its predictive power.

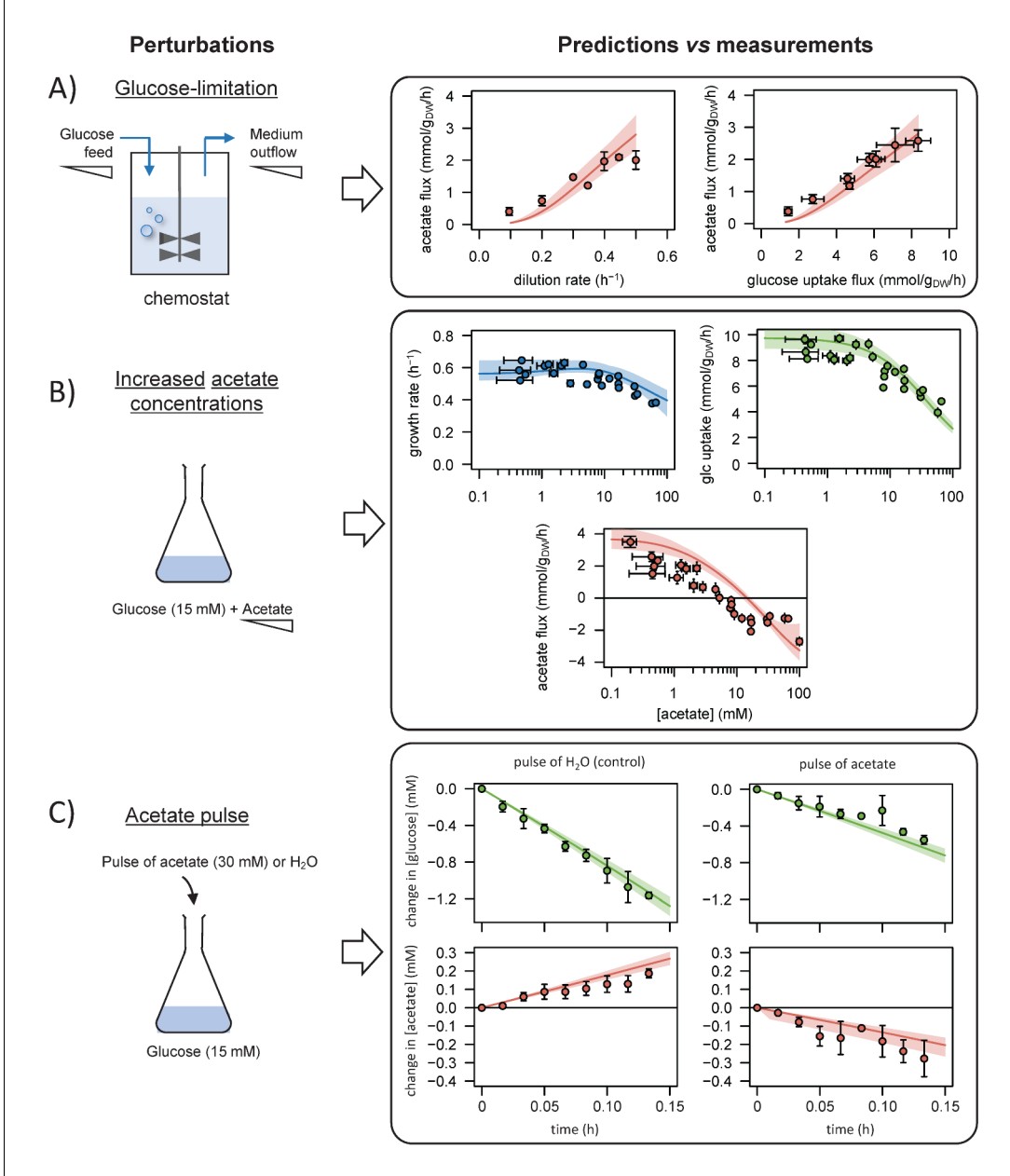

**Figure 3.** Comparison of model predictions with experimental data. We used the model to simulate (i) steady-state glucose and acetate fluxes in glucose-limited chemostat cultures at dilution rates of 0.1–0.5 hr$^{-1}$ (A), (ii) the growth rates and glucose and acetate fluxes during growth on glucose at various acetate concentrations (B), and (iii) the time courses of the changes in glucose and acetate concentrations during exponential growth on glucose after a pulse of either acetate or water (C). Model predictions are represented by lines and experimental data are shown as dots (the error bars represent one standard deviation), the shaded areas represent the 95% confidence intervals on the predictions. Predictions obtained with the alternative models (models 1–3) are shown in *Figure 3—figure supplements 1–3*. The predictive accuracy was compared between models based on the variance-weighted sum of squared residuals between simulated and experimental data (*Figure 3—figure supplements 1–4*).

The online version of this article includes the following source data and figure supplement(s) for figure 3:

**Source data 1.** Experimental data used to validate the model.
**Figure supplement 1.** Comparison of measured and predicted data (model 1).
**Figure supplement 2.** Comparison of measured and predicted data (model 2).
**Figure supplement 3.** Comparison of measured and predicted data (model 3).
**Figure supplement 4.** Comparison of the predictive accuracy of models 1–4.

To assess the functional importance of regulation by acetate, we predicted the response of *E. coli* to the same perturbations with alternative models (i.e., without inhibition of the glycolytic and/or TCA cycle pathways, models 1–3). The simulations in the absence of inhibition are qualitatively different (*Figure 3—figure supplements 1–3*). For instance, model 1 predicts a constant glucose uptake flux at all acetate concentrations, and even, contrary to observations, an increase in the growth rate with the acetate concentration. We compared the predictive accuracy of all the models by calculating the variance-weighted sum of squared residuals between the simulated and experimental data (*Figure 3—figure supplement 4*). The most accurate predictions for all perturbations are those from the model with dual inhibition (model 4), followed by those from the single-inhibition models (models 2 and 3) and finally those produced by the no-inhibition model (model 1). The predicted data are thus only qualitatively and quantitatively consistent with observations when inhibition by acetate is included, indicating that the regulatory role of acetate in *E. coli* is functionally important.

## Intracellular control of acetate flux is distributed around the acetyl-CoA node

We used a metabolic control analysis of this kinetic model to determine the degree of control exerted by each reaction on acetate flux (*Figure 4*). Flux control coefficients ($C_E^J$) quantify the impact of small changes in the rate of each reaction (typically from a change in the enzyme concentration $E$) on each flux ($J$) (*Kacser and Burns, 1973*; *Heinrich and Rapoport, 1974*).

Metabolic control analysis revealed that rather than being controlled entirely from within the acetate pathway, acetate flux is controlled to a similar degree by the acetate pathway (with a flux control coefficient of 0.76), glycolysis (0.88), and the TCA cycle (−0.64). The balance between acetyl-CoA production and demand (i.e., between glycolytic and TCA fluxes), therefore has a strong effect on acetate production. As expected, this control is positive for glycolysis (since it produces acetyl-CoA) and negative for the TCA cycle (since it consumes acetyl-CoA). Still, acetate flux is controlled to a large extent from within the acetate pathway, mainly by AckA (0.69), with a small contribution from acetate transport (0.07). Overall, acetate flux control is distributed around the acetyl-CoA node.

## Intracellular flux control patterns shift with the acetate concentration

Since control of acetate flux is distributed around the acetyl-CoA node, we tested how flux changes around this node might affect its control properties. Given that the concentration of acetate is a major determinant of acetate flux (*Figure 3*), we performed metabolic control analyses for a broad range of acetate concentrations (from 0.1 to 100 mM).

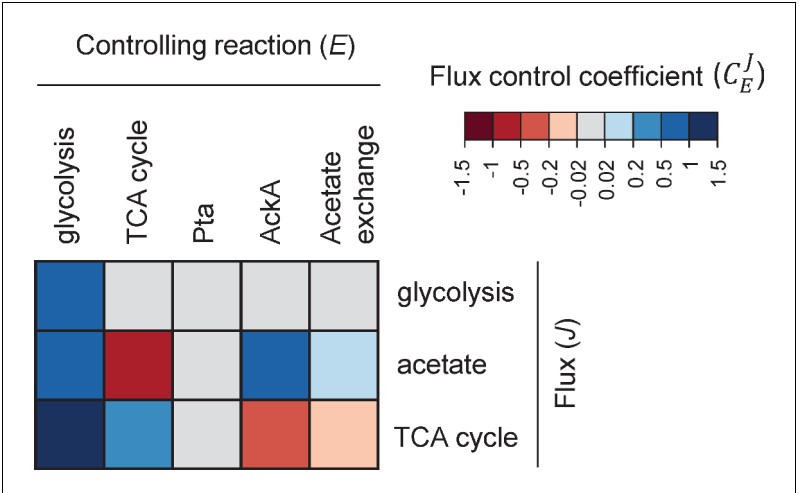

**Figure 4.** Heatmap of flux control coefficients during growth on glucose (15 mM) and acetate (0.1 mM). Each column represents a controlling reaction (E) and each row, a flux (J). Red and blue cells represent negative and positive flux control coefficients ($C_E^J$), respectively, with darker (lighter) tones indicating stronger (weaker) control.

The control exerted by the acetate pathway on acetate flux decreases non-linearly as the acetate concentration increases (*Figure 5A*), highlighting a progressive shift of the control from inside to outside the acetate pathway. This is contrary to the classical behaviour in which the control exerted by a metabolic pathway increases as it becomes saturated, i.e., where the flux reaches the capacity of the uptake pathway at high substrate concentrations (*Wegner et al., 2015*; *Moreno-Sánchez et al., 2008*). The control exerted by the acetate pathway gradually shifts towards the glycolytic and TCA pathways (*Figure 5B,C*). Discontinuous control patterns are observed for the latter pathways at a concentration of about 10 mM, i.e., around the concentration threshold at which the acetate flux switches from production to consumption and is zero (*Enjalbert et al., 2017*). The sum of the control coefficients of the glycolytic and TCA blocks, which represent the overall control exerted by *E. coli* metabolism on the acetate flux (*Figure 5D*), increases with the acetate concentration and compensates for the decrease in control from the acetate pathway. The ratio of the control coefficients of glycolysis and the TCA cycle show that control shifts from the glycolytic pathway to the TCA cycle as the acetate concentration increases (*Figure 5E*).

Overall, these results reveal how intracellular flux control patterns are strongly modulated by the extracellular concentration of acetate, with the control exerted by the acetate pathway being transferred progressively to the glycolytic and TCA pathways as the acetate concentration increases.

Here again, we tested the impact of regulation by acetate on the control properties of *E. coli* by comparing the results obtained with different models (*Figure 5—figure supplements 1–3*). The individual control profiles of the glycolytic and TCA pathways are qualitatively similar in all the models. However, the control profiles of the acetate pathway differ drastically in the absence of inhibition (model 1) and with partial inhibition (models 2 and 3), from the results with dual inhibition (model 4). In models 1–3, the control exerted by the acetate pathway remains constant at low to moderate

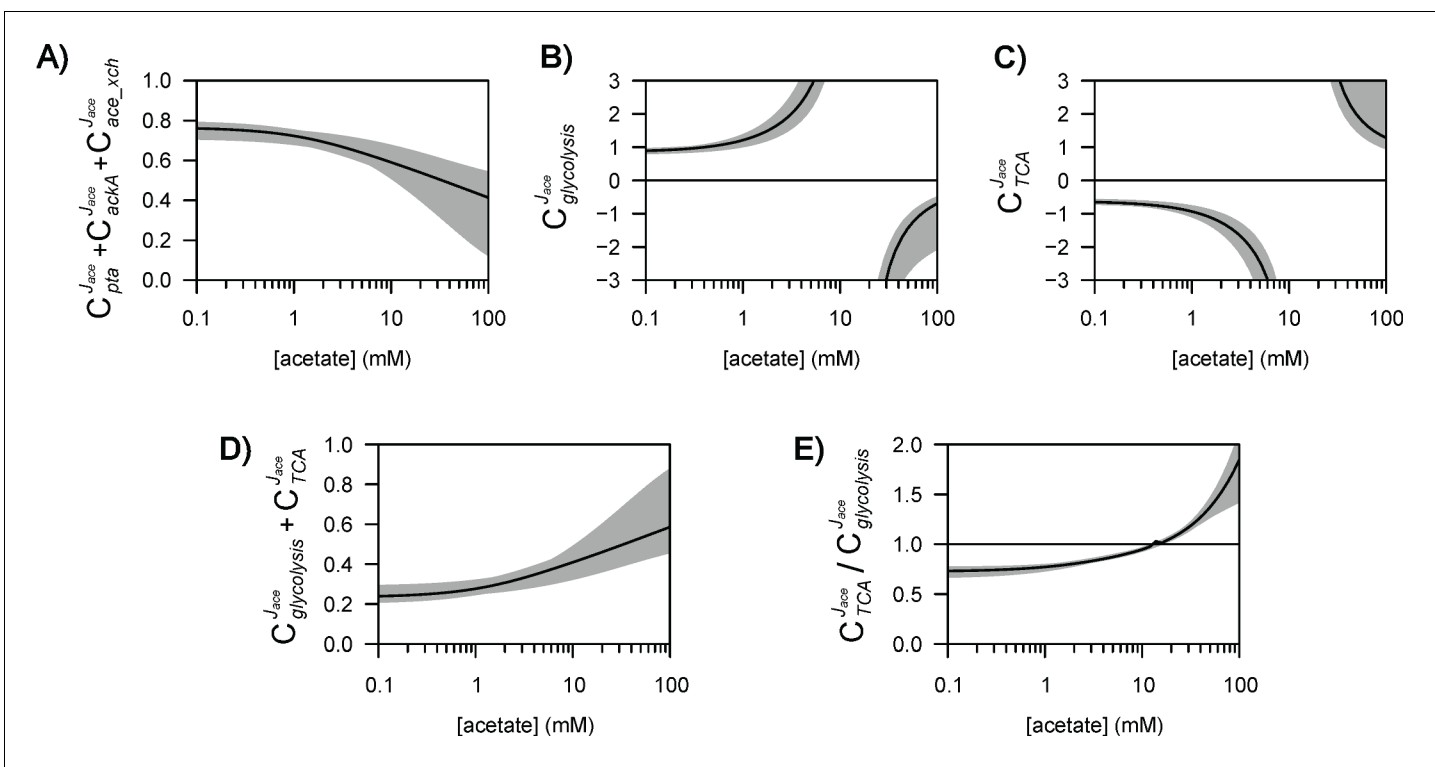

**Figure 5.** Control of acetate flux over a broad range of acetate concentrations. The shaded areas represent the 95% confidence intervals. Flux control coefficients calculated with the alternative models (models 1–3) are shown in *Figure 5—figure supplements 1–3*.

The online version of this article includes the following figure supplement(s) for figure 5:

**Figure supplement 1.** Flux control coefficients obtained with model 1.
**Figure supplement 2.** Flux control coefficients obtained with model 2.
**Figure supplement 3.** Flux control coefficients obtained with model 3.

acetate concentrations and tends to increase at high acetate concentrations, indicating that the progressive transfer of control from the acetate pathway to the rest of the metabolism as a function of the acetate concentration emerges specifically from the inhibition by acetate of both the glycolytic and the TCA cycle pathways.

## Regulation of *E. coli* metabolism by acetate

Metabolic control analysis identified the controlling steps, i.e., the reactions that alter the acetate flux if their rates are modified (e.g., through changes in enzyme concentrations). However, more information is needed to understand the mechanisms involved in flux responses to perturbations of an external parameter such as the acetate concentration. Indeed, while a metabolic reaction may exert some control on a given flux, this does not necessarily mean that it is involved in the observed flux response. For instance, AckA exerts some *control* on the acetate flux, but its expression appears to be constant at all acetate concentrations (*Figure 2*), hence it does not *regulate* the acetate flux at the transcriptional level.

To identify the regulatory routes that are actually involved in the response of *E. coli* to changes in acetate concentration, we used the concept of response coefficients ($R_{Ace}^{J_{ace}}$), which express the dependence of a system variable (here the acetate flux, $J_{ace}$) on an external effector (the concentration of acetate). The partitioned response relationship (*Kacser and Burns, 1973*; *Cornish-Bowden, 1995*) allows the flux response to a perturbation in acetate concentration channelled through a given reaction $i$ ($^{v_i}R_{Ace}^{J_{ace}}$) to be quantified, as detailed in the Materials and methods section. Since acetate regulates more than one reaction, the partitioned response coefficients provide a quantitative understanding of the different routes through which the acetate flux is regulated by the acetate concentration. We calculated the partitioned response coefficients of the acetate flux to the acetate concentration via (i) the acetate pathway, which represents the contribution of direct metabolic regulation, and via (ii) the glycolytic and (iii) the TCA pathways, where acetate acts indirectly by modulating the flux capacity (*Figure 6A*). As expected, regulation is minimal when acetate does not significantly modulate the acetate flux, i.e., at low and high acetate concentrations, and is maximal at concentrations that strongly modulate its flux (*Figure 6B–D*). Setting the response threshold of

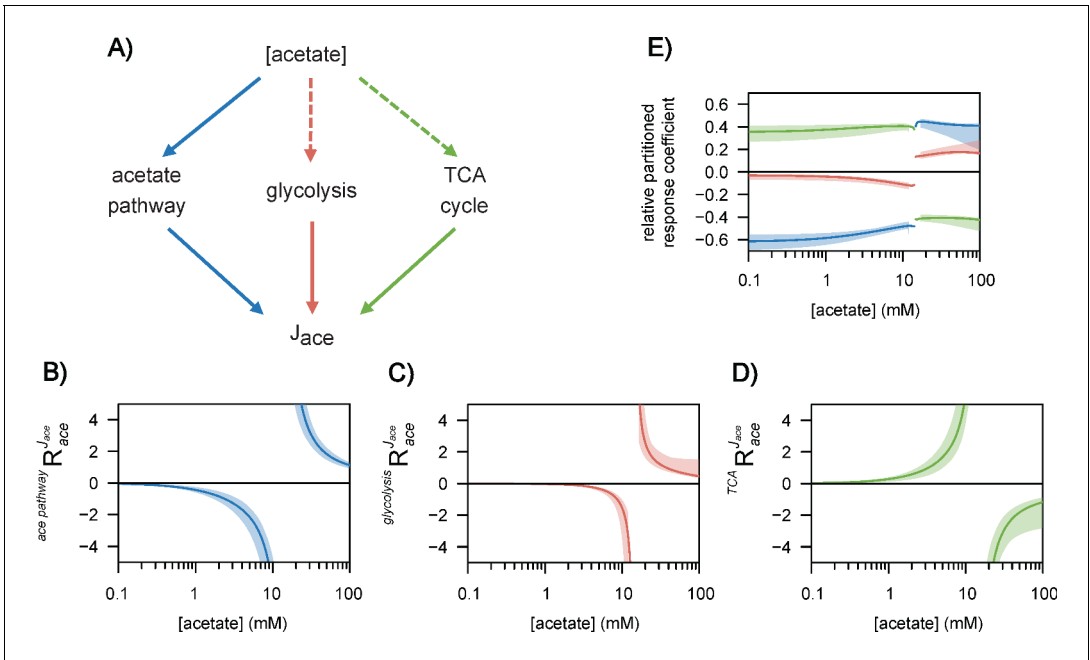

**Figure 6.** Regulation of acetate flux in *Escherichia coli*. The different routes through which acetate flux can be regulated by the acetate concentration are shown in (**A**). Dotted lines represent indirect (hierarchical) regulation, and straight lines represent direct (metabolic) regulation. The strengths of the three regulatory routes are respectively shown in (**B–D**), and their relative contributions are shown in (**E**). The shaded areas represent the 95% confidence intervals.

0.1 – i.e., that a relative change of $x$ % in the acetate concentration should lead to a relative flux response of at least $(0.1 \times x)$ % via the regulatory route considered – we determined that acetate acts as a regulator over a range of concentrations spanning three orders of magnitude (between 0.2 and 100 mM).

Based on the regulatory strength of each interaction, we determined the relative contribution of each interaction to the observed flux response (*Figure 6E*), revealing two distinct modes of regulation that depend on the acetate concentration and remain remarkably stable over a broad range of concentrations. At low acetate concentrations (<10 mM), the main regulatory route (50–60%) is direct metabolic regulation via the acetate pathway, with the remaining flux response being accounted for by indirect regulation through the TCA cycle (35–40%) and glycolysis (5–10%). This regulatory pattern remains stable before changing abruptly at ~10 mM when the acetate flux reverses. Above this threshold, direct regulation by acetate accounts for about 40% of the observed flux response, glycolysis, about 20%, and the TCA cycle, roughly 40%. This pattern then remains stable up to 100 mM.

These results provide a comprehensive and quantitative understanding of acetate flux regulation by acetate through the coordinated effects of direct and indirect mechanisms, with distinct regulatory programs at low and high acetate concentrations.

## Discussion

Two independent theories have been proposed to explain acetate overflow in *E. coli*, but neither global regulation of central metabolic pathways nor local control of the acetate pathway can account for all observations. Using a systems biology strategy, we reconcile these two theories, clarify how acetate flux is controlled and regulated, and establish the pivotal role acetate plays as a global regulator of glucose metabolism in *E. coli*.

The kinetic model of *E. coli* metabolism developed in this work links acetate metabolism with glucose uptake and growth. This model suggested the existence of a global regulatory program whereby the extracellular concentration of acetate determines the flux capacity of the glycolytic and TCA pathways, whose inhibition was required to explain the dynamic profiles obtained in $^{13}$C-labelling experiments. This inhibition was also necessary (and sufficient) to predict the steady-state and dynamic responses of *E. coli* to a broad range of perturbations. In this coarse-grained model, the simplified formalism used to consider acetate inhibition represents the integrated regulatory response of *E. coli* metabolism to acetate, regardless of the underlying molecular mechanisms. Regulation may indeed take place through a variety of mechanisms acting at the (post-)transcriptional, (post-)translational, and metabolic levels. These regulatory networks are tightly intertwined and act concertedly to ensure a coordinated response of *E. coli* to perturbations on the seconds to minutes time scale (*Enjalbert et al., 2013*; *Enjalbert et al., 2017*; *Chubukov et al., 2014*; *Lempp et al., 2019*; *von Wulffen et al., 2017*; *Gerosa and Sauer, 2011*). Transcriptomics experiments confirmed that acetate regulates the flux capacity of these two pathways at the transcriptional level by modulating the expression of most of their genes. This is a highly efficient way for cells to adjust their fluxes while maintaining metabolite homeostasis (*Kacser and Burns, 1973*; *Heinrich and Rapoport, 1974*). The metabolic response of *E. coli* to changes in acetate concentration thus involves a global reorganisation of its metabolism from the transcriptional to the flux levels, contrary to what has been suggested (*Pinhal et al., 2019*) and providing a mechanistic rationale for the reported 'toxicity' of acetate. The coordinated regulation of the glycolytic and TCA pathways cannot be explained by the actions of known transcriptional regulators, suggesting that these regulators act in concert or that there are additional regulators present.

Transcriptional regulation occurs in a few minutes (*Enjalbert et al., 2013*; *Lempp et al., 2019*; *von Wulffen et al., 2017*; *Morin et al., 2020*), which is sufficiently fast to explain most observations (*Figures 1* and *3A,B*). However, the immediate decrease in glucose uptake flux in response to a pulse of acetate (*Figure 3C*) suggests the existence of additional regulatory mechanisms that act on a faster time scale. These mechanisms could operate at the post-transcriptional level (possibly via the carbon storage regulator system [*Revelles et al., 2013*] or the BarA/UvrY two-component system [*Chavez et al., 2010*]), at the post-translational level (possibly via protein acetylation by acetyl-phosphate [*Weinert et al., 2013*; *Ren et al., 2019*]) and/or directly at the metabolic level itself (possibly via α-ketoglutarate [*Doucette et al., 2011*] or ATP [*Koebmann et al., 2002*], which control the

glycolytic flux). Further work will be required to identify the complete regulatory network of sense and response to acetate in *E. coli*. This work may be facilitated by the coarse grained model developed here as well as by more detailed kinetic models of *E. coli* metabolism (*Millard et al., 2017*; *Kadir et al., 2010*; *Jahan et al., 2016*). These models should help to computationally evaluate the biological relevance of detailed regulatory mechanisms, as illustrated here.

Our results confirm that the direction of the acetate flux is controlled thermodynamically and is fundamentally determined by the extracellular concentration of acetate. Moreover, we found that intracellular control of the acetate flux is distributed around the acetyl-CoA node, i.e., between the glycolytic, TCA, and acetate pathways, such that none of the three processes can be regarded as the sole kinetic bottleneck. The inhibition of the glycolytic and TCA pathways by acetate gives rise to unusual control properties. The intracellular flux control pattern is indeed determined by the concentration of acetate itself, with a progressive shift of control from the acetate pathway to the rest of the metabolism when the acetate concentration increases. This means that metabolic engineering interventions aiming at minimising acetate production in biotechnology should rebalance the flux capacity of both the glycolytic and the TCA pathways, possibly via global metabolic regulators, while accounting for the concentration-dependent effect of acetate on intracellular control patterns. The proposed model may thus be used to guide rational design of optimised strains for synthetic biology applications.

This study also identifies the regulatory scheme involved in *E. coli*'s response to changes in acetate concentration, which largely determines the acetate flux itself as well as the cell's physiology. We demonstrate that this scheme involves direct metabolic regulation of the acetate pathway in combination with indirect – at least partly transcriptional – regulation of the glycolytic and TCA pathways, and we show how this equal contribution of direct and indirect regulatory mechanisms explains the diverse patterns of acetate flux observed and forms part of *E. coli*'s global physiological response to changing acetate concentrations. The distinct regulatory patterns at high and low acetate concentrations also determine the metabolic status of acetate as a co-substrate or by-product of glucose metabolism. This work shows that the regulation of acetate flux is far more subtle than previously considered.

These results call for a reconsideration of the role of acetate, and more generally of overflow metabolism, in *E. coli*. Acetate should not be considered a terminal, wasteful product of glucose metabolism since it may equally be a co-substrate of glucose. Moreover, it is also a key regulator that induces a global remodelling of *E. coli* physiology and modulates several cellular functions (motility, biofilm formation, transport, metabolism, translation). Acetate regulates *E. coli* metabolism over a range of concentrations spanning three orders of magnitude, and its regulatory action involves a combination of direct and indirect mechanisms. In the laboratory, *E. coli* is typically grown on glucose at low acetate concentrations, which enhances its production. In the intestine however, *E. coli*'s environmental niche, the concentration of acetate is high (between 30 and 100 mM) (*Macfarlane et al., 1992*; *Cummings and Englyst, 1987*) and well above the ~10 mM threshold at which the acetate flux reverses. These levels of acetate should thus favour its co-utilisation with available sugars, suggesting that acetogenic species may be an important source of nutrients for *E. coli*. Besides its importance in terms of understanding cross-feeding relationships, this work also stresses the need to further investigate the regulatory role of acetate (and of acetogenic species) in the intestine. This will require a quantitative characterisation of the available nutrients and of their dynamics in this particularly complex environment.

Microorganisms other than *E. coli* are similarly capable of co-consuming glucose with its metabolic by-products, even when glucose is abundant. *Saccharomyces cerevisiae* can co-consume ethanol and glucose (*Raamsdonk et al., 2001*), and mammalian cells co-assimilate lactate with glucose (*Hui et al., 2020*; *Hui et al., 2017*; *Rabinowitz and Enerbäck, 2020*). These similarities, which have important implications for cell energy and redox balance, indicate that overflow metabolism should be considered a reversible process and point to the existence of a universal phenomenon similar to the one described here for acetate in *E. coli*. The quantitative systems biology approach developed in this work can be used as a guide for future investigations of overflow fluxes, their regulation, and their biological implications in *E. coli* and other (micro)organisms.

## Materials and methods

### Strain and conditions

*E. coli* K-12 MG1655 was grown in M9 medium (*Nicolas et al., 2007*) complemented with 15 mM glucose. Sodium acetate (prepared in a 1.6 M solution at pH 7.0) was added up to the required concentration. The cells were grown in shake flasks at 37°C and 200 rpm, in 50 mL of medium. For the isotope labelling experiments, unlabelled glucose was replaced by U-$^{13}$C$_6$-glucose (Eurisotop, France). Growth was monitored by measuring the optical density (OD) at 600 nm using a Genesys 6 spectrophotometer (Thermo, USA), and a conversion factor of 0.37 g$_{DW}$/L/OD unit (*Revelles et al., 2013*) was used to determine the biomass concentration.

### Transcriptomics experiments

Cells were grown in flasks in M9 minimal media with 15 mM glucose and 0, 10, 50, or 100 mM acetate. In mid-exponential growth phase (OD$_{600nm}$ = 1), 4 mL of each culture was centrifuged for 90 s at 14,000 rpm, the supernatant was discarded, and the pellets were immediately frozen in liquid nitrogen. Total RNA was extracted using a Qiagen RNAeasy MiniKit and quantified using a Nanodrop spectrophotometer. Double-stranded complementary DNA (cDNA) synthesis and array processing were performed by One-Color Microarray-Based gene Expression Analysis (Agilent Technologies). The images were analysed with the software DEVA (v1.2.1). All array procedures were performed using the GeT-Biopuces platform (http://get.genotoul.fr). Four independent biological replicates were analysed for each condition. For each data set, the log2 intensities obtained in the presence of acetate were divided by the log2 intensities obtained in the absence of acetate. These ratios were then normalised by the log2 median intensity. Genes whose expression level differed by a factor of 2 or more between the two conditions were used for further analysis. Gene ontology analyses were performed using Ecocyc (https://ecocyc.org), p-values were estimated using Fisher exact test with Bonferroni correction. The transcriptomics data can be downloaded from the ArrayExpress database (http://www.ebi.ac.uk/arrayexpress) under accession number E-MTAB-9086. Theoretical expression data at a given growth rate were obtained by extrapolating the data from *Esquerré et al., 2014*.

### Metabolomics experiments

Extracellular concentrations of labelled and unlabelled glucose and acetate were quantified during growth by 1D $^1$H-NMR on a Bruker Ascend 800 MHz spectrometer equipped with a 5 mm QCI cryoprobe (Bruker, Germany), as detailed previously (*Millard et al., 2014*). Briefly, 150 μL of filtered broth (0.2 μm, Sartorius, Germany) were mixed with 50 μL of D$_2$O containing TSPd4 (3-(trimethylsilyl)-1-propanesulfonic acid-tetra deuterated, used as internal standard) at a concentration of 10 mM. A sequence using presaturation (ZGPR) was used for water signal suppression, with a 30° pulse and a relaxation delay between scans of 10 s to ensure full signal recovery. A total of 32 scans were accumulated (64 k data points with a spectral width of 16 ppm) after 4 dummy scans. From each spectrum, we quantified glucose and unlabelled and labelled acetate. Spectra were processed using TopSpin v3.2 (Bruker).

### Model construction and analysis

All models are provided in SBML and COPASI formats in the Supplementary Information (*Supplementary file 1*) and at https://github.com/MetaSys-LISBP/acetate_regulation; *Millard, 2021*; copy archived at swh:1:rev:c8bcf8fad3459269df30dba7e52c81e62ca181d0. The calibrated kinetic model has also been deposited in the Biomodels database (https://www.ebi.ac.uk/biomodels) (*Chelliah et al., 2015*) with the identifier MODEL2005050001. Model analysis was performed using COPASI (v4.27, *Hoops et al., 2006*) with the *CoRC* package (COPASI R Connector v0.7.1, https://github.com/jpahle/CoRC, *Förster and Bergmann, 2021*) in R (v3.6.1, https://www.r-project.org). The scripts used to perform the simulations, to analyse the models and to generate the figures are provided in the Supplementary Information (*Supplementary file 1*) and at https://github.com/MetaSys-LISBP/acetate_regulation to ensure reproducibility and reusability.

## Model construction

The model contains six reactions, six species, and two compartments (the environment and the cell, with a cell volume of $1.77 \times 10^{-3}$ (L/g$_{DW}$) *Chassagnole et al., 2002*; *Figure 1A*). Model units are litre (L) for volumes, hour (hr) for time, and millimole (mmol), and gram dry weight (g$_{DW}$) for amounts of metabolites and biomass, respectively. All reactions are listed in *Table 1*. Glycolysis (which produces acetyl-CoA from glucose) and the TCA cycle (which utilises acetyl-CoA to produce biomass) were modelled using irreversible Michaelis-Menten kinetics. Growth rates were calculated from the flux of the TCA cycle assuming a constant biomass production yield from acetyl-CoA, in keeping with observations (*Enjalbert et al., 2017*; *Pinhal et al., 2019*). Acetate exchange between the cell and its environment was modelled as a saturable process using reversible Michaelis-Menten kinetics (*Millard et al., 2017*), and the Pta-AckA pathway was modelled using the detailed kinetics of the Pta and AckA enzymes used in previous models (*Enjalbert et al., 2017*; *Millard et al., 2017*; *Kadir et al., 2010*).

The differential equations, which describe the progression of the variables over time as a function of the system's rates, balance the concentrations of extracellular (biomass, glucose, and acetate) and intracellular (acetate, acetyl-CoA, and acetyl-phosphate) species:

$$\frac{dGLC}{dt} = v_{glycolysis} \cdot X \cdot \frac{V_{cell}}{V_{env}} \quad [+v_{feed} - D \cdot GLC]$$

$$\frac{dACE_{env}}{dt} = v_{acetate\_exchange} \cdot X \cdot \frac{V_{cell}}{V_{env}} \quad [-D \cdot ACE_{env}]$$

$$\frac{dX}{dt} = X \cdot v_{growth} \quad [-D \cdot X]$$

$$\frac{dACCOA}{dt} = 1.4 \cdot v_{glycolysis} - v_{TCA_{cycle}} - v_{Pta}$$

$$\frac{dACP}{dt} = v_{Pta} - v_{AckA}$$

$$\frac{dACE_{cell}}{dt} = v_{AckA} - v_{acetate\_exchange}$$

**Table 1.** Reactions included in the kinetic model of glucose and acetate metabolism of *Escherichia coli*.

| Name | Reaction | Rate law | Comment |
|---|---|---|---|
| Glucose_feed | ø → GLC | C | Glucose inflow and medium outflow to simulate chemostat experiments |
| Acetate_outflow | ACE$_{env}$ → ø | MA | |
| Biomass_outflow | X → ø | MA | |
| Glucose_outflow | GLC → ø | MA | |
| Glycolysis | GLC → 1.4 × ACCOA | IMM | Stoichiometric coefficient taken from *Millard et al., 2014* |
| TCA_cycle | ACCOA → ø | IMM | Utilisation of AcCoA by the TCA cycle |
| Pta | ACCOA ↔ ACP | RMM | Rate law from *Enjalbert et al., 2017*; *Millard et al., 2017*; *Kadir et al., 2010* |
| AckA | ACP ↔ ACE$_{cell}$ | RMM | Rate law from *Enjalbert et al., 2017*; *Millard et al., 2017*; *Kadir et al., 2010* |
| Acetate_exchange | ACE$_{cell}$ ↔ ACE$_{env}$ | RMM | Rate law from *Millard et al., 2017* |
| Growth | X → 2 × X | MA | Rate calculated from the TCA cycle flux, assuming a constant biomass yield (*Enjalbert et al., 2017*; *Pinhal et al., 2019*) |

Abbreviations: C: constant flux; MA: mass action; RMM: reversible Michaelis-Menten; IMM: irreversible Michaelis-Menten.

Terms within square brackets are required only to simulate chemostat experiments (at dilution rate $D$ and with a glucose feed defined by $v_{feed}$).

Reaction rates were modelled using the following rate laws:

$$v_{glycolysis} = \frac{Vmax_{glycolysis} \cdot GLC}{GLC + Km\_GLC} \left[ \cdot \frac{1}{1 + \frac{ACE_{env}}{Ki\_ACE}} \right]$$

$$v_{TCA\_cycle} = \frac{Vmax_{TCA\_cycle} \cdot ACCOA}{ACCOA + Km\_ACCOA} \left[ \cdot \frac{1}{1 + \frac{ACE_{env}}{Ki\_ACE}} \right]$$

$$v_{AckA} = \frac{\frac{Vmax_{AckA} \cdot \left( ACP \cdot ADP - \frac{ACE_{cell} \cdot ATP}{Keq} \right)}{Km\_ACP \cdot Km\_ADP}}{\left( 1 + \frac{ACP}{Km\_ACP} + \frac{ACE_{cell}}{Km\_ACE} \right) \cdot \left( 1 + \frac{ADP}{Km\_ADP} + \frac{ATP}{Km\_ATP} \right)}$$

$$v_{Pta} = \frac{\frac{Vmax_{Pta} \cdot \left( ACCOA \cdot P - \frac{ACP \cdot COA}{Keq} \right)}{Km\_ACCOA \cdot Km\_P}}{1 + \frac{ACCOA}{Km\_ACCOA} + \frac{P}{Ki\_P} + \frac{ACP}{Ki\_ACP} + \frac{COA}{Km\_COA} + \frac{ACCOA \cdot P}{Km\_ACCOA \cdot Km\_P} + \frac{ACP \cdot COA}{Km\_ACP \cdot Km\_COA}}$$

$$v_{Acetate\_exchange} = \frac{\frac{Vmax_{Acetate\_exchange} \cdot \left( ACE_{cell} - \frac{ACE_{env}}{Keq} \right)}{Km\_ACE}}{1 + \frac{ACE_{cell}}{Km\_ACE} + \frac{ACE_{env}}{Km\_ACE}}$$

$$v_{Growth} = v_{TCA\_cycle} \cdot Y$$

As detailed in the Results section, we constructed four different versions of this model, i.e., with or without (non-competitive) inhibition of the glycolytic and/or TCA cycle pathways by acetate (terms within square brackets). This regulatory term represents the immediate, integrated response of *E. coli* metabolism to changes in acetate concentration, with no a priori knowledge on the underlying molecular mechanisms. The controlling species is defined as being extracellular acetate, i.e., the initial environmental signal sensed by the cells. Similar results and conclusions are obtained if the controlling species is the intracellular acetate pool.

Finally, these models were extended with isotopic equations for parameter estimation, as detailed in *Enjalbert et al., 2017* and *Millard et al., 2015*. Briefly, all reactions (except biomass synthesis) were considered separately for unlabelled and labelled metabolites. Rate laws of reversible reactions were decomposed into their forward and reverse components to account for the bidirectional isotope exchange that arise from reversibility and significantly affects the distribution of isotopes through the network. For instance, the isotopically extended balance of acetyl-CoA corresponds to:

$$\frac{dACCOA_0}{dt} = \frac{GLC_0}{GLC_0 + GLC_1} \cdot 1.4 \cdot v_{glycolysis} + \frac{ACP_0}{ACP_0 + ACP_1} \cdot v_{Pta}^{reverse} - \frac{ACCOA_0}{ACCOA_0 + ACCOA_1} \cdot \left( v_{TCA_{cycle}} + v_{Pta}^{forward} \right)$$

$$\frac{dACCOA_1}{dt} = \frac{GLC_1}{GLC_0 + GLC_1} \cdot 1.4 \cdot v_{glycolysis} + \frac{ACP_1}{ACP_0 + ACP_1} \cdot v_{Pta}^{reverse} - \frac{ACCOA_1}{ACCOA_0 + ACCOA_1} \cdot \left( v_{TCA_{cycle}} + v_{Pta}^{forward} \right)$$

where subscripts 0 and 1 refer to the unlabelled and labelled metabolites, respectively.

## Concentration of cofactors

Concentrations of cofactors were taken from a published kinetic model of the Pta-AckA pathway (*Enjalbert et al., 2017*; ADP = 0.61 mM, ATP = 2.40 mM, CoA = 1.22 mM, P = 10 mM).

## Parameter estimation

The values of 14 of the 24 parameters were taken directly from the literature (*Table 2*). Parameters whose values are not available from elsewhere, which do not have a real biochemical value, or for which biochemical measurements are generally not representative of intracellular conditions (e.g.,

Vmax) were estimated to optimally reproduce 152 experimental data points obtained from *E. coli* K-12 MG1655 grown on $^{13}$C-glucose (15 mM) plus $^{12}$C-acetate (1, 10, or 30 mM). These data included time-course concentrations of biomass, glucose, and acetate and $^{13}$C-labelling of acetate. The parameters *p* were estimated by minimising the objective function *f* defined as the weighted sum of squared errors:

$$f(p) = \sum_i \left( \frac{x_i - y_i(p)}{\sigma_i} \right)^2$$

where $x_i$ is the experimental value of data point *i*, with an experimental standard deviation $\sigma_i$, and $y_i(p)$ is the corresponding simulated value. The objective function *f* was minimised with the particle swarm optimisation algorithm (2,000 iterations with a swarm size of 50). The experimental and fitted data are shown in *Figure 1* and provided in the Supplementary Information (*Supplementary file 1*). Values and 95% confidence interval of the estimated parameters are given in *Table 3*.

## Goodness-of-fit analysis

We used $\chi^2$ statistical tests to assess the goodness of fit of each model and determine whether they described the data with sufficient accuracy. The minimised variance-weighted sum of squared residuals (SSR) is a stochastic variable with a $\chi^2$ distribution. The acceptable threshold for SSR values is $\chi^2(\alpha, d)$, where *d* represents the number of degrees of freedom and is equal to the number of fitted measurements *n* minus the number of estimated independent parameters *p*. The parameter $\alpha$ was set to 0.95 to define a 95% confidence threshold and models with SSRs above this threshold were rejected since they cannot accurately reproduce the experimental data.

## Model validation

A total of 170 experimental data (extracellular fluxes and concentrations) were collected from the literature to evaluate the predictive power of each model. These data, which were *not* used to train or fit the models, were obtained for *E. coli* K-12 MG1655 and its close derivative strain BW25113 grown on different concentrations of glucose and/or acetate, as detailed in the Results section and in *Renilla et al., 2012*; *Enjalbert et al., 2017*; *Pinhal et al., 2019*. The steady-state and dynamic responses of *E. coli* to the corresponding perturbations were predicted using each model (*Figure 3*). The residual error (SSR) was calculated for each model from the simulated and experimental validation data (*Figure 3—figure supplement 4*) to identify the model that yielded the most accurate predictions.

**Table 2.** Values of kinetic parameters taken from the literature.

| Reaction | Parameter | Value | Source |
|---|---|---|---|
| AckA | Keq | 174 | *Enjalbert et al., 2017*; *Millard et al., 2017*; *Kadir et al., 2010* |
| | Km_ACE | 7 | *Enjalbert et al., 2017*; *Millard et al., 2017*; *Fox and Roseman, 1986* |
| | Km_ACP | 0.16 | *Enjalbert et al., 2017*; *Millard et al., 2017*; *Fox and Roseman, 1986* |
| | Km_ADP | 0.5 | *Enjalbert et al., 2017*; *Millard et al., 2017*; *Fox and Roseman, 1986* |
| | Km_ATP | 0.07 | *Enjalbert et al., 2017*; *Millard et al., 2017*; *Fox and Roseman, 1986* |
| Pta | Keq | 0.005 | *Enjalbert et al., 2017*; *Millard et al., 2017* |
| | Km_ACCOA | 0.2 | *Enjalbert et al., 2017*; *Millard et al., 2017*; *Campos-Bermudez et al., 2010* |
| | Ki_ACP | 0.2 | *Enjalbert et al., 2017*; *Millard et al., 2017*; *Campos-Bermudez et al., 2010* |
| | Km_COA | 0.029 | *Enjalbert et al., 2017*; *Millard et al., 2017*; *Campos-Bermudez et al., 2010* |
| | Ki_P | 13.5 | *Enjalbert et al., 2017*; *Millard et al., 2017*; *Campos-Bermudez et al., 2010* |
| | Km_ACP | 0.7 | *Enjalbert et al., 2017*; *Millard et al., 2017*; *Campos-Bermudez et al., 2010* |
| | Km_P | 6.1 | *Enjalbert et al., 2017*; *Millard et al., 2017* |
| Glycolysis | Km_GLC | 0.02 | *Jahreis et al., 2008* |
| Acetate exchange | Keq | 1 | Intracellular and extracellular concentrations equilibrate over time |

**Table 3.** Values and 95% confidence intervals of the estimated parameters.

| Reaction | Parameter | Value | 95 % CI |
|---|---|---|---|
| AckA | Vmax | $3.4 \times 10^5$ | $2.8 \times 10^5 - 5.5 \times 10^5$ |
| Pta | Vmax | $9.8 \times 10^5$ | $4.9 \times 10^4 - 9.9 \times 10^6$ |
| Glycolysis | Vmax | $5.6 \times 10^3$ | $5.3 \times 10^3 - 5.9 \times 10^3$ |
|  | Ki_ACE | 36.7 | 30.9 – 46.9 |
| TCA cycle | Km_ACCOA | 24.8 | 8.4 – 615.4 |
|  | Vmax | $7.4 \times 10^5$ | $2.4 \times 10^5 - 1.7 \times 10^6$ |
|  | Ki_ACE | 2.3 | 1.8 – 3.4 |
| Growth | Y | $1.0 \times 10^{-4}$ | $9 \times 10^{-5} - 1.1 \times 10^{-4}$ |
| Acetate exchange | Vmax | $4.8 \times 10^5$ | $8 \times 10^4 - 1.5 \times 10^6$ |
|  | Km_ACE | 33.2 | 1.5 – 99.8 |

## Metabolic control and regulation analyses

Scaled flux control coefficients ($C_E^J$), which represent the fractional change in the steady-state flux $J$ in response to a fractional change in the rate of the reaction $E$ (*Kacser and Burns, 1973*; *Heinrich and Rapoport, 1974*; *Cornish-Bowden, 1995*), were calculated as follows:

$$C_E^J = \frac{\partial lnJ}{\partial lnE}$$

Similarly, we defined the response coefficient ($R_{Ace}^{J_{ace}}$) which represents the dependence of the acetate flux ($J_{ace}$) on the extracellular concentration of acetate (*Kacser and Burns, 1973*; *Cornish-Bowden, 1995*):

$$R_{Ace}^{J_{ace}} = \frac{\partial lnJ_{ace}}{\partial lnAce}$$

The partitioned response relationship (*Kacser and Burns, 1973*; *Cornish-Bowden, 1995*) was used to quantify the flux response ($^{v_i}R_{Ace}^{J_{ace}}$) to a change in acetate concentration channelled through reaction $i$. The effect of the acetate concentration on the rate of reaction $i$ ($v_i$) is described by the elasticity coefficient $\varepsilon_{Ace}^{v_i}$, and the resulting change in $v_i$ then propagates through the system depending on the control exerted by reaction $i$ on the acetate flux ($C_{v_i}^{J_{ace}}$):

$$^{v_i}R_{Ace}^{J_{ace}} = C_{v_i}^{J_{ace}} \cdot \varepsilon_{Ace}^{v_i}$$

with

$$\varepsilon_{Ace}^{v_i} = \frac{\partial lnv_i}{\partial lnAce}$$

## Global sensitivity analyses

We used a Monte-Carlo approach (*Saa and Nielsen, 2017*) to determine the 95% confidence intervals on (i) the fits of the experimental data (*Figure 1*), (ii) the estimated parameters (*Table 3*), (iii) the predicted responses to perturbations (*Figure 3*), and (iv) the flux control and regulation coefficients (*Figures 5* and *6*). For this purpose, we generated 500 simulated sets of calibration data with noise added according to experimental standard deviations. For each of these artificially noisy data sets, we carried out complete computational analyses (i.e., including parameter estimation – starting from random initial parameter values, simulation of the responses to different perturbations, and metabolic control and regulation analyses). We calculated the mean value and 95% confidence intervals around each parameter and each variable (i.e., concentrations, fluxes, flux control coefficients, and response coefficients) from the distribution of values obtained for the 500 data sets.

## Acknowledgements

The authors thank MetaboHub-MetaToul (Metabolomics and Fluxomics facilities, Toulouse, France, http://www.metatoul.fr), which is part of the French National Infrastructure for Metabolomics and Fluxomics (http://www.metabohub.fr/), funded by the ANR (MetaboHUB-ANR-11-INBS-0010), for access to NMR facilities. JCP is grateful for funding from INSERM for his temporary full-time researcher position. The authors are grateful for Vincent Pierunek's help with the isotope labelling experiments, and wish to thank the following INSA Toulouse students for help with the transcriptomics experiments: Leidy Caraballo, Xavier Caron, Pauline Chanut, Sarah Guiziou, Ngoc Thu Hang Pham, Diane Barbay, Céline Ben Hassen, Thomas Cerutti, Cécile Roland, Sarah Srour, Audrey Baylet, Mathilde Beraud, Claudie Bosc, Lilas Courtot, Violaine Dolfo, Anna Kaci, Manon Chevallot-Beroux, Sarah Colom, Fanny Leclerc, Zihan Liao, Grégoire Quinet and Mélina Vaurs.

## Additional information

### Funding

| Funder | Grant reference number | Author |
| --- | --- | --- |
| Institut National de Recherche pour l'Agriculture, l'Alimentation et l'Environnement | MICA-JC | Pierre Millard |

The funders had no role in study design, data collection and interpretation, or the decision to submit the work for publication.

### Author contributions

Pierre Millard, Conceptualization, Software, Formal analysis, Supervision, Funding acquisition, Investigation, Visualization, Methodology, Writing - original draft, Project administration, Writing - review and editing; Brice Enjalbert, Formal analysis, Investigation, Visualization, Methodology, Writing - review and editing; Sandrine Uttenweiler-Joseph, Investigation, Writing - review and editing; Jean-Charles Portais, Fabien Létisse, Funding acquisition, Writing - review and editing

### Author ORCIDs

Pierre Millard https://orcid.org/0000-0002-8136-9963
Brice Enjalbert https://orcid.org/0000-0003-1291-1373
Sandrine Uttenweiler-Joseph https://orcid.org/0000-0001-9019-4766
Jean-Charles Portais https://orcid.org/0000-0002-3480-0933
Fabien Létisse https://orcid.org/0000-0002-1490-0152

### Decision letter and Author response

Decision letter https://doi.org/10.7554/eLife.63661.sa1
Author response https://doi.org/10.7554/eLife.63661.sa2

## Additional files

### Supplementary files

• Supplementary file 1. R scripts used to construct the models, perform the simulations and generate the figures.

• Transparent reporting form

### Data availability

Transcriptomics data have been deposited in ArrayExpress under accession code E-MTAB-9086. The calibrated kinetic model has been deposited in BioModels database under accession code MODEL2005050001. All the scripts used to perform the simulations, to analyse the models and to generate the figures are provided in the supporting files and at https://github.com/MetaSys-LISBP/

acetate_regulation (copy archived https://archive.softwareheritage.org/swh:1:rev: c8bcf8fad3459269df30dba7e52c81e62ca181d0/). All data generated or analysed during this study are included in the manuscript and supporting files.

The following datasets were generated:

| Author(s) | Year | Dataset title | Dataset URL | Database and Identifier |
|---|---|---|---|---|
| Millard P, Enjalbert B, Uttenweiler-Joseph S, Portais JC, Letisse F | 2020 | Response of *Escherichia coli* to acetate concentration during growth on glucose | https://www.ebi.ac.uk/arrayexpress/experiments/E-MTAB-9086/ | ArrayExpress, E-MTAB-9086 |
| Millard P | 2020 | Calibrated kinetic model of glucose and acetate metabolisms of *Escherichia coli* | https://www.ebi.ac.uk/biomodels/MODEL2005050001 | BioModels, MODEL2005050001 |

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
