## [Decision Letter]

**Acceptance summary:**

This paper provides an analysis of the metabolic feedback mechanisms regulating overflow metabolism during glycolysis in *E. coli*. The authors present a kinetic metabolic model incorporating glycolysis, the TCA cycle, and acetate metabolism, and argue that acetate needs to suppress glycolysis and TCA cycle activity for the model to be able to reproduce experimental findings. The assumption of acetate activity is further validated through experiments showing that acetate reduces expression of the glucose phosphotransferase system and of most TCA cycle genes. Another finding is that the control of acetate production shifts from inside (low acetate concentrations) to outside (high concentrations) the acetate pathways.

**Decision letter after peer review:**

Thank you for submitting your article "Control and regulation of acetate overflow in *Escherichia coli*" for consideration by *eLife*. Your article has been reviewed by three peer reviewers, one of whom is a member of our Board of Reviewing Editors, and the evaluation has been overseen by Naama Barkai as the Senior Editor. The reviewers have opted to remain anonymous.

The reviewers have discussed the reviews with one another and the Reviewing Editor has drafted this decision to help you prepare a revised submission.

As the editors have judged that your manuscript is of interest, but as described below that additional work is required before it can be published, we would like to draw your attention to changes in our revision policy that we have made in response to COVID-19 (https://elifesciences.org/articles/57162). First, because many researchers have temporarily lost access to the labs, we will give authors as much time as they need to submit revised manuscripts. We are also offering, if you choose, to post the manuscript to bioRxiv (if it is not already there) along with this decision letter and a formal designation that the manuscript is "in revision at *eLife*". Please let us know if you would like to pursue this option. (If your work is more suitable for medRxiv, you will need to post the preprint yourself, as the mechanisms for us to do so are still in development.)

Essential revisions:

1) Modeling choices, model robustness and the question of alternative models need to be more thoroughly addressed.

2) Interpretation of experimental results need to be strengthened, in particular with regards to the time scales of transcription and regulation.

3) All comments made by reviewers need to be thoroughly and constructively addressed.

Reviewer #2 (Recommendations for the authors (required)):

The manuscript delves into a major classical question in the study of metabolism, namely how organisms balance the decision of respiring or fermenting carbon sources, and what role fermentation by-products play in this process. Specifically, the authors focus on *E. coli* and the dependence of its physiology on acetate present in the environment. This is an interesting topic relevant to a broad spectrum of questions, including cancer metabolism. The approach is original, and the specific hypothesis that acetate influences transcription and the metabolic flows in previously unknown ways is well-posed and intriguing. I must admit though that I am not totally convinced that the results provide conclusive evidence about their hypothesis, ruling out other possibilities. Clear comparisons with alternative models, parameter sensitivity analyses, and a clearer expression of possible caveats would alleviate this concern, and help readers clearly understand the possible limitations of the approach employed. The work should be of interest to a fairly broad audience of biochemists, microbiologists and systems biologists.

Subsection “Testing the model”: the authors mention an excellent agreement between model and experiment in Figure 3. I think the agreement is not bad, but I would not call it excellent. I am not sure a p-value is absolutely necessary, but it would be good to at least have an idea of how much variability the experimental data may have beyond the shown error bars, and whether the agreement is really unexpected, also in light of the parameter fitting presented in Figure 1.

It would be useful to see Figure 3 in absence of the acetate inhibition effects. Would the results look only quantitatively different or also qualitatively different? Can the authors comment on the intuitive meaning of these curves. Is any of these curves dramatically affected by the new model, relative to alternative standard models?

As opposed to what is mentioned in the Introduction, stoichiometric models could in principle help address the questions posed by the authors. This would not be the case for regular FBA models, but would be true for dynamics FBA models, in which the extracellular concentration of substrates is explicitly accounted for.

The “Metabolomics experiments” subsection in the Materials and methods is very brief, and does not get into the detail of flux measurements reported, which are crucial for the model and its interpretation.

I am curious about whether acetate inflow and outflow are separately measurable (and measured) in the system, or whether the reports are for the net flow. I am not sure if multiple acetate transporters exist, and whether it is in principle possible for *E. coli* to simultaneously secrete acetate due to fermentation and take it up from the environment, but it would be good for the authors to comment on this.

No parameter sensitivity analysis was performed. I would recommend that the authors either add it, or comment on why it would not be useful.

The authors' conclusions are presented in an absolute way, that leaves no room for alternative explanations. Partly due to the issues mentioned above, after reading the paper I felt highly intrigued by the results, but not 100% convinced that their explanation is the only possible. I would suggest that the authors (a) take steps to present in a more critical way the results in light of other models that were not explicitly tested, and (b) acknowledge explicitly possible limitation of this study, and avenues to further corroborate the proposed mechanisms.

Introduction, second paragraph: it becomes clear from the context, but the authors should clearly mention that they refer to extracellular concentration of acetate.

Reviewer #3 (Recommendations for the authors (required)):

The manuscript presents an extension of previous work of the group (Enjalbert et al., 2017; Millard, Smallbone and Mendes, 2017). The authors present an interesting dynamical model of overflow metabolism, producing new insights into a long-standing question in bacterial physiology. The model is confronted with new data that appear to confirm specific model predictions. The major novelty of the model is the addition of a direct control by acetate of the activities of glycolysis (or glucose uptake) and the TCA cycle. Very interesting transcriptomics data appear to confirm these regulatory relationships. However, transcriptional regulation and the direct regulations suggested by the model act at different time-scales. In conclusion, the work is very well carried out and interesting. The connection between the transcriptomics data and the model need to be clarified.

1) The first major argument in favor of control of glycolysis and the TCA cycle by acetate is the quality of fit to bioreactor data. It does not seem surprising to me that the quality of fit (Figure 1B) is improved by adding additional regulatory interactions. Even though the figure shows a significant improvement of the fits, it would be nice to see the alternative fits (the equivalent of Figure 1C) in supplementary information.

2) A strong claim of the manuscript is that a kinetic model is needed to account for experimental observations. The fits in Figure 1 use the dynamical model at steady state. How do the predictions compare to the alternative models (global control of metabolism, local control of the acetate pathway)?

3) The transcriptomics data are very interesting. However, these data were acquired at steady state, reflecting a slow adaptation (order of magnitude of the generation time) of physiology. The regulatory interactions (square brackets in the equations in the subsection “Model construction”) are instantaneous effects of the concentration of acetate. I fail to see how the transcriptomics data can argue in favor (or disfavor) of the kinetic model.

4) The perturbation experiments (Figure 3) are probably the best test of a dynamical model. The model correctly predicts the observed dynamics, which are clearly on time-scale of minutes, precluding transcriptional regulation as a cause. The authors could suggest possible molecular mechanisms.

5) The remaining results of the manuscript address model properties and are very well carried out. Under the hypothesis that the model is correct, these analyses yield interesting insights into the physiology of overflow metabolism.

In conclusion: the manuscript describes a very interesting extension to current models of overflow metabolism in *E. coli*. The experiments are very well carried out and pertinent new data are presented.

My major concern is that the transcriptomics data seem disconnected from the kinetic model. Long-term adaptation can be achieved by modifications in gene expression. However, this does not correspond to the mechanism described by the model equations. The manuscript would be much stronger if the authors could show at least one of the molecular mechanisms by which acetate controls metabolic pathways. I realize that this would be "another" manuscript and cannot be reasonably covered by major revisions. The authors could at least discuss possible molecular mechanisms of this regulation.

---

## [Author Response]

Essential revisions:1) Modeling choices, model robustness and the question of alternative models need to be more thoroughly addressed.

In the revised manuscript, we provide additional details to clarify our modelling choices, in particular regarding (i) the formalism used to model the regulation of metabolism by acetate and (ii) the definition of extracellular acetate as the controlling factor. We have also extensively tested the robustness of the model by performing global sensitivity analyses for each step, from model construction to the analysis of regulation, which confirm that the model and its predictions are robust to parameter uncertainty. Finally, simulations performed with alternative models highlight the biological importance of the identified regulatory interaction on the properties of the system. We appreciate these constructive suggestions, which significantly improve the manuscript.

2) Interpretation of experimental results need to be strengthened, in particular with regards to the time scales of transcription and regulation.

We have strengthened the interpretation of the experimental results, in particular with regards to the time scales of the different regulatory mechanisms.

3) All comments made by reviewers need to be thoroughly and constructively addressed.

All of the reviewers’ comments are thoroughly addressed below.

Reviewer #2 (Recommendations for the authors (required)):The manuscript delves into a major classical question in the study of metabolism, namely how organisms balance the decision of respiring or fermenting carbon sources, and what role fermentation by-products play in this process. Specifically, the authors focus on *E. coli* and the dependence of its physiology on acetate present in the environment. This is an interesting topic relevant to a broad spectrum of questions, including cancer metabolism. The approach is original, and the specific hypothesis that acetate influences transcription and the metabolic flows in previously unknown ways is well-posed and intriguing. I must admit though that I am not totally convinced that the results provide conclusive evidence about their hypothesis, ruling out other possibilities. Clear comparisons with alternative models, parameter sensitivity analyses, and a clearer expression of possible caveats would alleviate this concern, and help readers clearly understand the possible limitations of the approach employed. The work should be of interest to a fairly broad audience of biochemists, microbiologists and systems biologists.

We thank reviewer 2 for their insightful comments and constructive suggestions.

The revised manuscript contains detailed comparisons with alternative models (i.e. models without inhibition of the glycolytic and/or TCA cycle pathways by acetate), which highlight the importance of the identified regulatory interaction for the functioning of the system and its control properties. We have also performed a global sensitivity analysis, which confirms our conclusions are robust to parameter uncertainty, and we discuss the limitations of our approach.

We would like to stress that the simplified formalism used to consider acetate inhibition represents the integrated regulatory response of *E. coli* metabolism to acetate, without any a priori knowledge of the exact mechanism. Experimental results confirm that acetate does indeed regulate gene expression in *E. coli*, but this does not rule out other possibilities. In fact, as stated in the original manuscript, we believe that other mechanisms work in concert, and that acetate may “also regulate *E. coli* metabolism at the post-translational level (e.g. via the carbon storage regulator system [Revelles et al., 2013] or acetyl-phosphate [Weinert et al., 2013; Ren et al., 2019]) and at the metabolic level (e.g. via the control of glucose uptake by α-ketoglutarate [Doucette et al., 2011] or of glycolysis by ATP [Koebmann et al., 2002])”. These regulatory mechanisms are likely also involved in the response of *E. coli* to acetate, and this should be explored in future work. We have made this point clearer in the revised manuscript.

Subsection “Testing the model”: the authors mention an excellent agreement between model and experiment in Figure 3. I think the agreement is not bad, but I would not call it excellent. I am not sure a p-value is absolutely necessary, but it would be good to at least have an idea of how much variability the experimental data may have beyond the shown error bars, and whether the agreement is really unexpected, also in light of the parameter fitting presented in Figure 1.

We agree with this recommendation. In the revised manuscript, we have removed the term “excellent”, and have carried out global sensitivity analyses to determine 95% confidence intervals around the model predictions. We now evaluate these predictions based on a quantitative metric, namely the variance-weighted sum of squared residuals between predicted and experimental data. This metric is useful to compare the proposed model with alternatives. Results indicate that the model with inhibitions, which is already the only model that satisfactorily fits the data shown in Figure 1, is also the one that most accurately predicts the data used for validation (Figure 3). We present and discuss these new results in the revised manuscript (in the Results and Discussion sections).

It would be useful to see Figure 3 in absence of the acetate inhibition effects. Would the results look only quantitatively different or also qualitatively different? Can the authors comment on the intuitive meaning of these curves. Is any of these curves dramatically affected by the new model, relative to alternative standard models?

We have followed this suggestion and have predicted the responses to the perturbations shown in Figure 3 with the alternative models, i.e. in the absence of acetate inhibition of the glycolytic and/or TCA pathways (see Figure 3—figure supplements 1–4). The new simulation results are quantitatively and qualitatively different, which confirms that the curves are drastically affected by the absence of inhibition, further supporting the regulatory role of acetate. We have also carried out metabolic control analyses for the alternative models (see Figure 5—figure supplements 1–3), which also indicate that the control properties of the system are qualitatively affected by the absence of inhibition. These new results are discussed in the revised manuscript.

As opposed to what is mentioned in the Introduction, stoichiometric models could in principle help address the questions posed by the authors. This would not be the case for regular FBA models, but would be true for dynamics FBA models, in which the extracellular concentration of substrates is explicitly accounted for.

We agree with reviewer 2 that dynamic FBA models can be used to estimate and predict dynamic flux reorganizations. However, dynamic FBA models are not purely stoichiometric since they include a kinetic description of the uptake pathways, and thus can equally be considered (partly) kinetic models.

One of the main objectives of this work was to better understand the control and regulation of acetate metabolism. We could have used dynamic FBA models to predict the response of *E. coli* to perturbations, but not to obtain a quantitative, mechanistic understanding of overflow metabolism. Indeed, to the best of our knowledge, dynamic FBA models lack regulation of intracellular pathways (such as the inhibition of the TCA cycle and of glycolysis identified in this work and implemented in our model) and do not account for phenomena such as enzyme saturation. Since these models cannot be used to compute elasticity coefficients for the intracellular pathways, they cannot be used either to quantify the flux control exerted by each pathway, or to determine the regulatory routes activated in response to changes of acetate concentration.

The “Metabolomics experiments” subsection in the Materials and methods is very brief, and does not get into the detail of flux measurements reported, which are crucial for the model and its interpretation.

The set of 152 experimental data used for model calibration consist only of time-course exometabolome concentrations and labellings, no flux data (though this model can be used to simulate such data, as shown in Figure 3). All the flux measurements reported in this study were obtained from the literature. As detailed in the Materials and methods section, to validate the model we used “A total of 170 experimental data (extracellular fluxes and concentrations) obtained from the literature for *E. coli* K-12 MG1655 and its close derivative strain BW25113 grown on glucose with or without acetate [Renilla et al., 2012; Enjalbert et al., 2017; Pinhal et al., 2019]”. The methods used to obtain these flux measurements are described in detail in the corresponding publications. We have made this point clearer in the revised manuscript:

“A total of 170 experimental data (extracellular fluxes and concentrations) were collected from the literature to evaluate the predictive power of each model. […] The steady-state and dynamic responses of *E. coli* to the corresponding perturbations were predicted using each model (Figure 3).”

We have also added the following details in the “Metabolomics experiments” subsection of the Materials and methods section:

“Briefly, 150 µL of filtered broth (0.2 μm, Sartorius, Germany) were mixed with 50 µL of D_2_O containing TSPd4 (3-(trimethylsilyl)-1-propanesulfonic acid-tetra deuterated, used as internal standard) at a concentration of 10 mM. […] Spectra were processed using TopSpin 3.2 (Bruker).”

I am curious about whether acetate inflow and outflow are separately measurable (and measured) in the system, or whether the reports are for the net flow. I am not sure if multiple acetate transporters exist, and whether it is in principle possible for *E. coli* to simultaneously secrete acetate due to fermentation and take it up from the environment, but it would be good for the authors to comment on this.

The dynamic ^13^C-labeling experiments performed in this work can indeed be exploited to determine acetate production and consumption fluxes separately. In fact, in a previous study (Enjalbert et al., 2017), we integrated these experiments into an isotopic (stoichiometric) model to demonstrate, as suggested by reviewer 2, that *E. coli* simultaneously produces acetate from glucose and takes it up from the environment. We have added a couple of sentences about these results in the revised manuscript:

“These experiments were designed to demonstrate that *E. coli* simultaneously produces acetate from glucose and consumes it from the environment [Enjalbert et al., 2017]. They provide information on the forward and reverse fluxes of acetate between the cell and its environment [Enjalbert et al., 2017], and thereby improve the calibration of the model.”

No parameter sensitivity analysis was performed. I would recommend that the authors either add it, or comment on why it would not be useful.

We agree, and are grateful for this suggestion. In the revised manuscript, we present the results of a global sensitivity analysis (Monte Carlo approach) to determine 95 % confidence intervals on (i) the fit of the experimental data (Figure 1), (ii) the estimated parameters (Table 3), (iii) the predicted responses to perturbations (Figure 3), and (iv) the flux control and regulation coefficients (Figures 5 and 6). These procedures are detailed in the revised manuscript. The results confirm that our conclusions are robust to parameter uncertainty.

The authors' conclusions are presented in an absolute way, that leaves no room for alternative explanations. Partly due to the issues mentioned above, after reading the paper I felt highly intrigued by the results, but not 100% convinced that their explanation is the only possible. I would suggest that the authors (a) take steps to present in a more critical way the results in light of other models that were not explicitly tested, and (b) acknowledge explicitly possible limitation of this study, and avenues to further corroborate the proposed mechanisms.

In the revised manuscript, we provide a more critical presentation of the results in light of other models that are now explicitly tested, we discuss the limitations of the approach, and we discuss alternative hypotheses and avenues to further investigate the regulatory role of acetate.

Introduction, second paragraph: it becomes clear from the context, but the authors should clearly mention that they refer to extracellular concentration of acetate.

We now refer explicitly to the extracellular concentration of acetate.

Reviewer #3 (Recommendations for the authors (required)):The manuscript presents an extension of previous work of the group (Enjalbert et al., 2017; Millard, Smallbone and Mendes, 2017). The authors present an interesting dynamical model of overflow metabolism, producing new insights into a long-standing question in bacterial physiology. The model is confronted with new data that appear to confirm specific model predictions. The major novelty of the model is the addition of a direct control by acetate of the activities of glycolysis (or glucose uptake) and the TCA cycle. Very interesting transcriptomics data appear to confirm these regulatory relationships. However, transcriptional regulation and the direct regulations suggested by the model act at different time-scales. In conclusion, the work is very well carried out and interesting. The connection between the transcriptomics data and the model need to be clarified.

We thank reviewer 3 for their positive feedback and for raising important points that we address and discuss in the revised manuscript. We have clarified the fact that regulation as encoded in the model does not aim to specifically represent transcriptional regulation (which is in fact fast, see our detailed response below) but rather the integrated response of *E. coli* to acetate. Transcriptomics data confirm that acetate regulates *E. coli* metabolism at least partly at the transcriptional level, and this does not exclude the possibility that other mechanisms may be involved in this integrated response. New analyses with the alternative models confirm that the proposed regulatory interactions are required not only to fit the data shown in Figure 1, but also to accurately predict the response of *E. coli* to perturbations (Figure 3 and Figure 3—figure supplements 1-4). Finally, we also demonstrate that unexpected control properties emerge from regulation by acetate (Figure 5 and Figure 5—figure supplements 1-3). The results of a global sensitivity analysis (suggested by reviewer 2) also strengthen these conclusions.

1) The first major argument in favor of control of glycolysis and the TCA cycle by acetate is the quality of fit to bioreactor data. It does not seem surprising to me that the quality of fit (Figure 1B) is improved by adding additional regulatory interactions. Even though the figure shows a significant improvement of the fits, it would be nice to see the alternative fits (the equivalent of Figure 1C) in supplementary information.

We have followed this suggestion and now provide alternative fits (equivalent to Figure 1C) in Figure 1—figure supplements 1–3. These new figures indeed help to visualize the extent to which the alternative models do not reproduce the data, as confirmed by the goodness-of-fit statistical analysis (Figure 1B).

2) A strong claim of the manuscript is that a kinetic model is needed to account for experimental observations. The fits in Figure 1 use the dynamical model at steady state. How do the predictions compare to the alternative models (global control of metabolism, local control of the acetate pathway)?

The fits in Figure 1 are not based on the dynamic model at steady-state. The simulated and experimental data are the time-course dynamics of biomass, glucose and acetate concentrations and of acetate labelling. We now provide the predictions of alternative models (responses to perturbations shown in Figure 3, as well as the control properties shown in Figure 5) in supplementary figures. The predictions are drastically affected by the absence of inhibition, confirming that regulation by acetate is necessary to explain the observed behaviours.

3) The transcriptomics data are very interesting. However, these data were acquired at steady state, reflecting a slow adaptation (order of magnitude of the generation time) of physiology. The regulatory interactions (square brackets in the equations in the subsection “Model construction”) are instantaneous effects of the concentration of acetate. I fail to see how the transcriptomics data can argue in favor (or disfavor) of the kinetic model.

As noted by reviewer 3, the inhibition of glycolysis and of the TCA cycle by acetate is considered immediate.

However, transcriptional regulation acts much faster than assumed by reviewer 3 and is not of the “order of magnitude of the generation time”. For instance, Kresnowati et al. (doi: 10.1038/msb4100083) found that glucose-limited *Saccharomyces cerevisiae* cells exhibit a major transcriptional reprogramming within the first 5 minutes after a glucose pulse. The authors showed that most TCA genes “were immediately downregulated” after the pulse. A similar time-scale has been reported for mRNA dynamics in *E. coli*, with a median mRNA half-live of just 3.1 min during exponential growth on glucose (Morin et al., 2020). This fast mRNA dynamics explains how the transcriptome of *E. coli* cells respond in less than 5 minutes to external perturbations, such as glucose exhaustion (Enjalbert, Letisse and Portais, 2013; Lempp et al., 2019) and changing oxygen levels (von Wulffen et al., 2017). As shown in the latter study, “significant differential expression of several genes within 30 seconds indicates immediate cell adaptation to the changed environment”. These findings indicate that the metabolic and transcriptional responses of *E. coli* and other microorganisms are tightly intertwined rather than occurring sequentially on different, independent time scales.

It seems important here to clarify that when constructing the model, the aim was not to specifically represent transcriptional regulation but rather the integrated response of *E. coli* to acetate. With this objective in mind, the model indeed provides new insights on global rules by which acetate governs *E. coli* metabolism. Transcriptomics experiments confirm that acetate control regulates the flux capacity of the glycolytic and TCA pathways at the transcriptional level, but we do not rule out other hypotheses. As discussed in the manuscript, we also believe that transcriptional regulation is not the only mechanism by which acetate regulates *E. coli* metabolism and physiology. We stated in the initial manuscript that acetate may “also regulate *E. coli* metabolism at the post-translational level (e.g. via the carbon storage regulator system [Revelles et al., 2013] or acetyl-phosphate [Weinert et al., 2013; Ren et al., 2019]) and at the metabolic level (e.g. via the control of glucose uptake by α-ketoglutarate [Doucette et al., 2011] or of glycolysis by ATP [Koebmann et al., 2002])”. These mechanisms act on a similar time scale – seconds to minutes – and are also represented by the feedback interaction included in the model. We have made this point clearer in the revised manuscript. Further work will be necessary to determine whether additional mechanisms are involved in the response of *E. coli* metabolism to acetate. We thank reviewer 3 for raising this important point, which we have clarified in the revised manuscript by adding the following sentences in the Discussion:

“In this coarse-grained model, the simplified formalism used to consider acetate inhibition represents the integrated regulatory response of *E. coli* metabolism to acetate, regardless of the underlying molecular mechanisms. […] These models should help to computationally evaluate the biological relevance of detailed regulatory mechanisms, as illustrated here.”

4) The perturbation experiments (Figure 3) are probably the best test of a dynamical model. The model correctly predicts the observed dynamics, which are clearly on time-scale of minutes, precluding transcriptional regulation as a cause. The authors could suggest possible molecular mechanisms.

As noted by reviewer 3, the perturbation experiments are indeed the best test of a dynamic model. The dynamics observed in response to an acetate pulse (Figure 3C) occur on a time-scale of 8 minutes, which cannot be explained by transcriptional regulation alone. In fact, as discussed in the manuscript, we also think it likely that acetate regulates *E. coli* metabolism via other mechanisms operating at the post-translational level or at the metabolic level. We have added a paragraph discussing in detail the time-scales of regulation and other possible regulatory mechanisms in the revised manuscript (see our answer to comment #3).

5) The remaining results of the manuscript address model properties and are very well carried out. Under the hypothesis that the model is correct, these analyses yield interesting insights into the physiology of overflow metabolism.

We thank reviewer 3 for this positive feedback on the analyses performed with the model.

In conclusion: the manuscript describes a very interesting extension to current models of overflow metabolism in *E. coli*. The experiments are very well carried out and pertinent new data are presented.My major concern is that the transcriptomics data seem disconnected from the kinetic model. Long-term adaptation can be achieved by modifications in gene expression. However, this does not correspond to the mechanism described by the model equations. The manuscript would be much stronger if the authors could show at least one of the molecular mechanisms by which acetate controls metabolic pathways. I realize that this would be "another" manuscript and cannot be reasonably covered by major revisions. The authors could at least discuss possible molecular mechanisms of this regulation.

We thank reviewer 3 for their constructive suggestions. Using the model, we determined that acetate inhibits the glycolytic and TCA fluxes, and identified its global impact on *E. coli* cells. Importantly, we show that this inhibition involves at least the transcriptomics response, which is in fact fast and tightly intertwined with the metabolic response (see above). We do not rule out other possible explanations and are working on identifying additional mechanisms involved in this response, which are indeed beyond the scope of this manuscript. In the revised manuscript, we discuss other molecular mechanisms possibly involved in the previously unknown regulatory role of acetate, and how the present model can help to address this question.